# Continuous high-fat high-sugar diet overrides the therapeutic potential of fecal microbiota transplantation from exercised and/or inulin-conditioned donors in obese mice

**Masato Kawashima**[1,2]*, **Takafumi Aoki**[3], **Hiroki Hamada**[2], **Chihiro Watanabe**[2], **Eri Oyanagi**[2], **Takumi Kono**[4], **Takashi Yamagata**[2], **Nicholas P. West**[5], **Hiromi Yano**[2,4]

**1** Department of Humanities and Social Sciences, Institute of Liberal Arts, Kyushu Institute of Technology, Kitakyushu, Japan, **2** Department of Health and Sports Science, Kawasaki University of Medical Welfare, Kurashiki, Japan, **3** Department of Clinical Nutrition, Kawasaki University of Medical Welfare, Kurashiki, Japan, **4** Graduate School of Health Science and Technology, Kawasaki University of Medical Welfare, Kurashiki, Japan, **5** School of Pharmacy and Medical Science and Central Facility for Genomics, Griffith University, Parklands Drive, QLD, Australia

* kawashima.masato623@mail.kyutech.jp



## Abstract

Fecal microbiota transplantation (FMT) is a promising therapeutic strategy for obesity and related metabolic disorders. Exercise and dietary fiber intake, such as inulin supplementation, have been shown to differentially modulate the gut microbiota and synergistically improve metabolic health. The present study aimed to investigate whether FMT from lean donor mice subjected to voluntary exercise and/or inulin supplementation could ameliorate metabolic dysfunction in high-fat high-sugar diet (HFHSD)-induced obese mice. Four-week-old male C57BL/6J mice were fed HFHSD throughout the experimental period and assigned to one of five groups: sham FMT, FMT from sedentary donors, from exercised donors, from inulin-supplemented donors, or from donors receiving both interventions. Following 12 weeks of obesity induction, mice were treated with antibiotics and then underwent a 4-week FMT protocol. Physical and metabolic parameters, gut microbial composition, and cecal short-chain fatty acid (SCFA) levels were examined in both donors and recipients. The results demonstrated that FMT from exercised and/or inulin-supplemented donors failed to improve obesity-related phenotypes or glucose intolerance in recipients. These outcomes were accompanied by only partial alterations in gut microbiota and SCFA profiles. Collectively, our findings suggest that persistent HFHSD exposure compromises the colonization and function of beneficial microbes, limiting the metabolic benefits of FMT. Successful application of FMT in severe obesity may require prior optimization of the host intestinal environment through dietary interventions or microbiome-targeted strategies.








**Data availability statement:** All relevant data are within the paper and its Supporting Information files.

**Funding:** This study was supported by Japan Society for the Promotion of Science (JSPS) KAKENHI Grant Numbers JP21K21270 (to M.K.) (URL: https://kaken.nii.ac.jp/en/grant/KAKENHI-PROJECT-21K21270/) and JP23K16738 (to M.K.) (URL: https://kaken.nii.ac.jp/en/grant/KAKENHI-PROJECT-23K16738/), and The Ryobiteien Memorial Foundation (URL: https://ryobi.gr.jp/teienzaidan/index.html) (to M.K.). The funders had no role in study design, data collection and analysis, decision to publish, or preparation of the manuscript.

**Competing interests:** The authors have declared that no competing interests exist.

## Introduction

Obesity and its associated metabolic co-morbidities, such as type 2 diabetes, cardiovascular disease, and metabolic dysfunction-associated steatotic liver disease (MASLD), pose major global health challenges [1–3]. Among emerging treatment options, fecal microbiota transplantation (FMT) has gained considerable attention for its potential to modulate host metabolism through gut microbial manipulation [4,5]. Advances in microbiome research have demonstrated that commensal microbes play a pivotal role in regulating host metabolic health [6,7], and that FMT can transfer donor-derived metabolic phenotypes to recipients [e.g., 8–10]. However, the clinical efficacy of FMT in managing obesity and related metabolic dysfunction remains inconsistent, particularly in individuals with severe obesity [11,12].

Diet and physical activity—two fundamental components of lifestyle intervention—are known to influence gut microbial composition [13,14]. In particular, habitual consumption of inulin, a commonly used prebiotic fiber, has been shown to modulate gut microbial communities and enhance the production of short-chain fatty acids (SCFAs), which are associated with improved metabolic profiles [15,16]. Notably, Rodriguez et al. (2022) recently reported that the combination of exercise and inulin supplementation produced synergistic effects on gut microbiota composition and glucose homeostasis [17]. These findings raise the possibility that FMT derived from donors exposed to both exercise and inulin may offer a novel and potentially more effective therapeutic strategy for obesity and its metabolic complications.

The purpose of the present study was to evaluate the impact of FMT from lean donor mice subjected to voluntary exercise and/or inulin supplementation on metabolic outcomes in obese recipient mice continuously fed a high-fat diet (HFD) and sugar-enriched water, a dietary regimen known to exacerbate glucose intolerance and insulin resistance [18].

## Materials and methods

### Animals and experimental design

Four-week-old male C57BL/6J mice (n = 84, CLEA Japan, Inc., Tokyo, Japan) were used in this study. The mice were randomly assigned to four donor groups and five recipient groups described below, and housed under a controlled environment (22 °C ± 1 °C, 12:12-h light/dark cycle) with free access to food and water throughout the experiments. Body weight and food intake were recorded weekly. The experimental period spanned a total of 17 weeks, including 12 weeks of obesity induction with high-fat high-sugar diet (HFHSD), one week of antibiotic treatment, and four weeks of FMT (Fig 1). This study was approved by the Institutional Animal Care and Use Committee of Kawasaki University of Medical Welfare (approval number: 21−008).

Donor mice (n = 24) were assigned the sedentary (Sed), exercise (Ex), Sed plus inulin (Sed + Inu), and Ex plus inulin (Ex + Inu) groups. In the Sed and Sed + Inu groups, mice were co-housed (four mice per cage) while in the Ex and Ex + Inu groups they were housed individually in a cage with a rotating wheel (ENV-044, Med Associates, Fairfax, VT, USA) allowing them to run freely. In the Ex and Ex + Inu groups, the number of revolutions of the running wheel was calculated in counts/

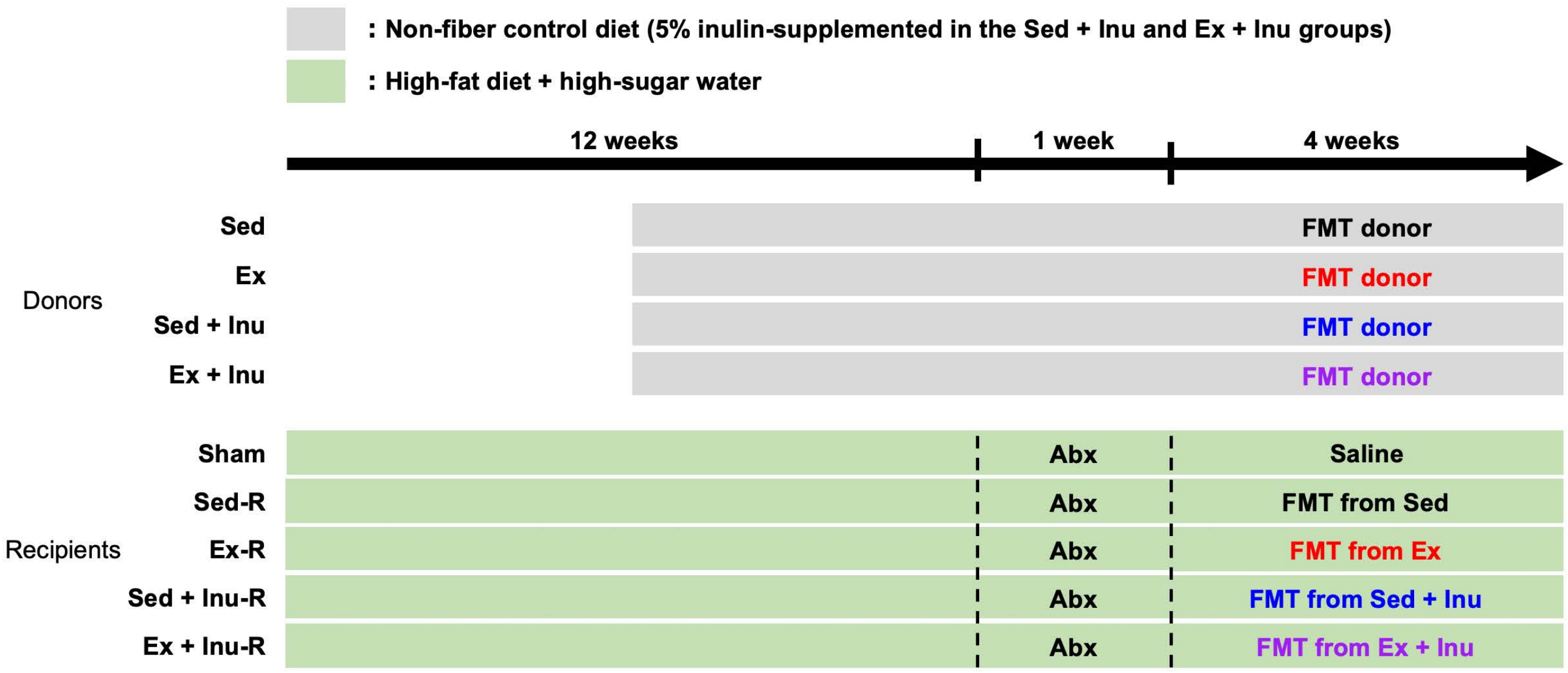


**Fig 1. Schematic design of the experiment.** After the 12-week-feeding with HFHSD, all recipient mice were given Abx for 1 week, followed by FMT from respective donors for 4 weeks. Abx: antibiotics, FMT: fecal microbiota transplantation, HFHSD: high-fat high-sugar diet.


week. The donor mice in the Sed and Ex groups were fed with a non-fiber diet (D13053007M, Research Diets, New Brunswick, NJ, USA) containing 10% fat, 20% protein, and 70% carbohydrate of total calories, and in the Sed + Inu and Ex + Inu groups mice were fed with the same diet supplemented an inulin (5% in weight, I2255, Sigma-Aldrich, St. Louis, MO, USA) throughout the experimental period.

All recipient mice (n = 60) were co-housed (four per cage) and fed the HFHSD for 12 weeks to induce obesity. The HFHSD was composed of the HFD (D12492, Research Diets), which provided 60% fat, 20% protein, and 20% carbohydrate by total caloric content, and sugary water supplemented with 42 g/L of sugar (55% fructose and 45% sucrose) [19,20]. To record the amount of individual food intake, the recipient mice were moved to individual cages at the 11th week of the feeding period. After another week of individual housing, the mice were allocated to one of the five groups: sham FMT (Sham), FMT from donors in the Sed group (Sed-R), FMT from donors in the Ex group (Ex-R), FMT from donors in the Sed + Inu group (Sed + Inu-R), and FMT from donors in the Ex + Inu group (Ex + Inu-R); with the adjustment to ensure that the mean body weights across all recipient groups are equal. Afterward, the antibiotic treatment and FMT were performed as described in the following sections.

### Antibiotic treatment

All mice in the five recipient groups were placed on broad spectrum antibiotics with ampicillin (1 mg/ml) and neomycin (0.5 mg/ml) (Sigma-Aldrich) in drinking water for one week as previously reported [21–23]. As ampicillin and neomycin are poorly absorbed, the treatment primarily influences intestinal microbiota without direct systemic effects [22,24].

### The procedure of FMT

The FMT to recipient mice was conducted daily for 4 weeks (28 days) following antibiotic treatment, as described in our previous report [23]. Briefly, fresh feces from donor mice were collected and pooled by group every day, and 5–7 fecal pellets (100–150 mg) were suspended in sterile saline at a concentration of 100 mg/ml. The supernatant was obtained after a short centrifugation. The crude aqueous fecal extract from donors was administered to the respective recipient mice

via oral gavage (100 µl each). In the Sham group, mice were subjected to the same paradigm as other recipient groups, except that mice were gavaged with 100 µl of saline.

### Glucose tolerance test (GTT)

Before and after the FMT, GTTs were performed following 6 h-fasting. A small amount of blood was collected from the tail vein, and blood glucose levels were measured using a glucose monitoring device Accu-Chek (Roche, Basel, Switzerland) at rest (0), 15, 30, 60, and 120 min after glucose administration (2 g/kg body weight, i.p.). Each mouse was lightly anesthetized with isoflurane inhalation prior to the glucose administration.

### Sample collection and blood analysis

All mice were euthanized by cervical dislocation at the end of experimental period and samples were collected. White adipose tissue (epididymal fat), large intestine, and cecum were dissected and weighed. The cecal content was immediately frozen in liquid nitrogen. Plasma was prepared from blood samples by centrifugation (3000 rpm, 20 min, 4 °C), then stored at −80 °C. The plasma samples were delivered to SRL, Inc. (Tokyo, Japan), and the levels of triglyceride, total cholesterol, free fat acid (FFA), aspartate aminotransferase (AST), and alanine aminotransferase (ALT) were measured.

### Real-time quantitative polymerase chain reaction (RT-qPCR)

Total RNA was extracted using TRIzol Reagent (Invitrogen, Carlsbad, CA) and RNeasy Mini Kit (Qiagen, Valencia, CA). RNA purity was assessed using the NanoDrop One (Thermo Fisher Scientific, Inc., Waltham, MA). RT-qPCR was performed using the reverse transcription kit (High-Capacity cDNA Reverse Transcription Kit; Thermo Fisher Scientific) and the step one plus real-time PCR system (Applied Biosystems) with Fast SYBR Green Master Mix (Applied Biosystems). The following amplification procedure was applied: initial denaturation for 10 min at 95 °C, followed by 40 cycles of denaturation for 3 s at 95 °C and annealing for 15 s at 60 °C. Glycer-aldehyde-3-phosphate dehydrogenase (Gapdh) mRNA was used as the housekeeping gene, and all data are represented relative to its expression, using standard curve methods. The specific PCR primer pair for the targeted genes are shown in Table 1.

### Gut microbiota analysis

Collected fresh fecal samples were immediately frozen in liquid nitrogen and stored at −80°C. Bacterial DNA was extracted from the fecal samples using a QIAamp Fast Stool DNA Mini Kit (Qiagen). The extracted bacterial DNA was then subjected to amplicon sequence analysis using the MiSeq system (Illumina, San Diego, CA, USA) by TechnoSuruga Laboratory Co., Ltd. (Shizuoka, Japan). DNA was extracted using an automated DNA isolation system (GENEPRE PSTARPI-480, Kurabo, Osaka, Japan), with 200 µL of distilled water included as a negative control sample.

The V3–V4 regions of Prokaryote 16S rRNA were amplified from the extracted DNA using the Pro341F/Pro805R primers and the dual index method. A negative control sample was also included, and the amplicons were visualized by electrophoresis.

Pro341F: 5′-CCTACGGGNBGCASCAG-3′

**Table 1. Primer sequences.**

| Primer | | Sequence (5’ – 3’) |
|---|---|---|
| *Tnf-α* | F | CCTCCCTCTCATCAGTTCTA |
| | R | ACTTGGTGGTTTGCTACGAC |
| *Il-1β* | F | AAAAAAGCCTCGTGCTGTCG |
| | R | GTCGTTGCTTGGTTCTCCTTG |


Pro805R: 5′-GACTACNVGGGTATCTAATCC-3′

Barcoded amplicons were paired-end sequenced on a 2 × 284 bp cycle using the MiSeq system with MiSeq Reagent Kit v3 (15 Gb: 600 Cycle). The quality of the paired-end sequencing reads was checked using the FASTX-Toolkit, and they were merged using the fastq-join program with the default settings. The joined reads extracted a quality value score (QC) of ≥ 20 for more than 99% of the sequence. The sequences' homology of ≥ 97% identity was clustered into the same bacterial species identification using the Metagenome@KIN Ver 2.2.1 analysis software (World Fusion, Tokyo, Japan) and the TechnoSuruga Lab Microbial Identification database DB-BA 13.0 (TechnoSuruga Laboratory). The 16S rRNA data were analyzed with Quantitative In-sights Into Microbial Ecology (QIIME) 2.0 ver. 2023.9 [25]. Quality filtering and chimeric sequences were filtered using DADA2 (Divisive Amplicon Denoising Algorithm 2) de-noise-single plugin ver. 2017.6.0 with the default option [26]. Taxonomic assignment of amplicon sequence variants (ASVs) was performed using the Silva database ver. 138 based on an average percent identity of 99% [27]. For taxonomic diversity, the α-diversity was analyzed using the Chao 1 [28], Shannon [29], and Simpson [30] indices. Differences in β-diversity were visualized using principal coordinates analysis (PCoA) based on unweighted UniFrac distance. Linear discriminant analysis (LDA) effect size (LEfSe) analysis was used to determine the significantly enriched taxa in each group [31]. In LEfSe analysis, the LDA score was computed for taxa differentially abundant between the different groups and top 5 genera/species in each group were obtained.

### Analysis of SCFAs (acetate, propionate, and butyrate) in the cecum

The frozen cecal samples from donors and recipients were transferred to TechnoSuruga Laboratory Co., Ltd., and the concentrations of acetate, propionate, and butyrate were determined by high-performance liquid chromatography (HPLC) (Prominence; Shimadzu, Kyoto, Japan) with a CDD-10Avp conductivity detector (Shimadzu, Kyoto, Japan), tandemly arranged in two columns (Shim-pack SCR-102(H); 300 mm × 8.0 mm ID), and a guard column and a Shim-pack SCR-102(H) guard column (50 mm × 6.0 mm ID) [32].

### Statistical analysis

Data are expressed as the mean ± standard error of the mean (SEM) or beeswarm boxplots. The Shapiro-Wilk test and Levene's test was used for confirming the normality and equal variance. For the comparison among donors or recipient groups, data with normality and equal variance were analyzed using one-way analysis of variance (ANOVA) followed by Bonferroni's post hoc test. Non-normality and/or unequal variance data were analyzed using the Kruskal-Wallis test followed by Steel-Dwass's post hoc test. The statistical distances in gut microbiota β-diversity were analyzed with permutational multivariate analysis of variance (PERMANOVA) using R Studio (ver. 2023.06.1 + 524) with the vegan package using the ASV-level Unifrac distance. P-values of $< 0.05$ were considered statistically significant. Statistical analyses were performed using R software (version 4.3.1; R Foundation for Statistical Computing Platform, Vienna, Austria). The underlying data for all figures are provided in the Supporting Information (S1 Appendix).

## Results

### Physical characteristics and glucose metabolism in donor mice

The voluntary running activity of donor mice in the Ex and Ex + Inu groups remained comparable throughout the experimental period (Ex group: 41,069.1 ± 4,202.3 counts/week; Ex + Inu group: 41,406.3 ± 3,126.4 counts/week; mean ± SEM). At the end of the experiment, body weights in the exercised (Ex and Ex + Inu) groups were significantly lower than those in the non-exercised (Sed and Sed + Inu) groups (Fig 2A, $P < 0.01$). Furthermore, body weight in the Sed + Inu group was significantly higher than in the Sed group (Fig 2A, $P < 0.05$). Additionally, each intervention (Ex, Sed + Inu, and Ex + Inu) significantly reduced body fat compared to the sedentary control (Fig 2B, $P < 0.01$). Notably, the combination of exercise

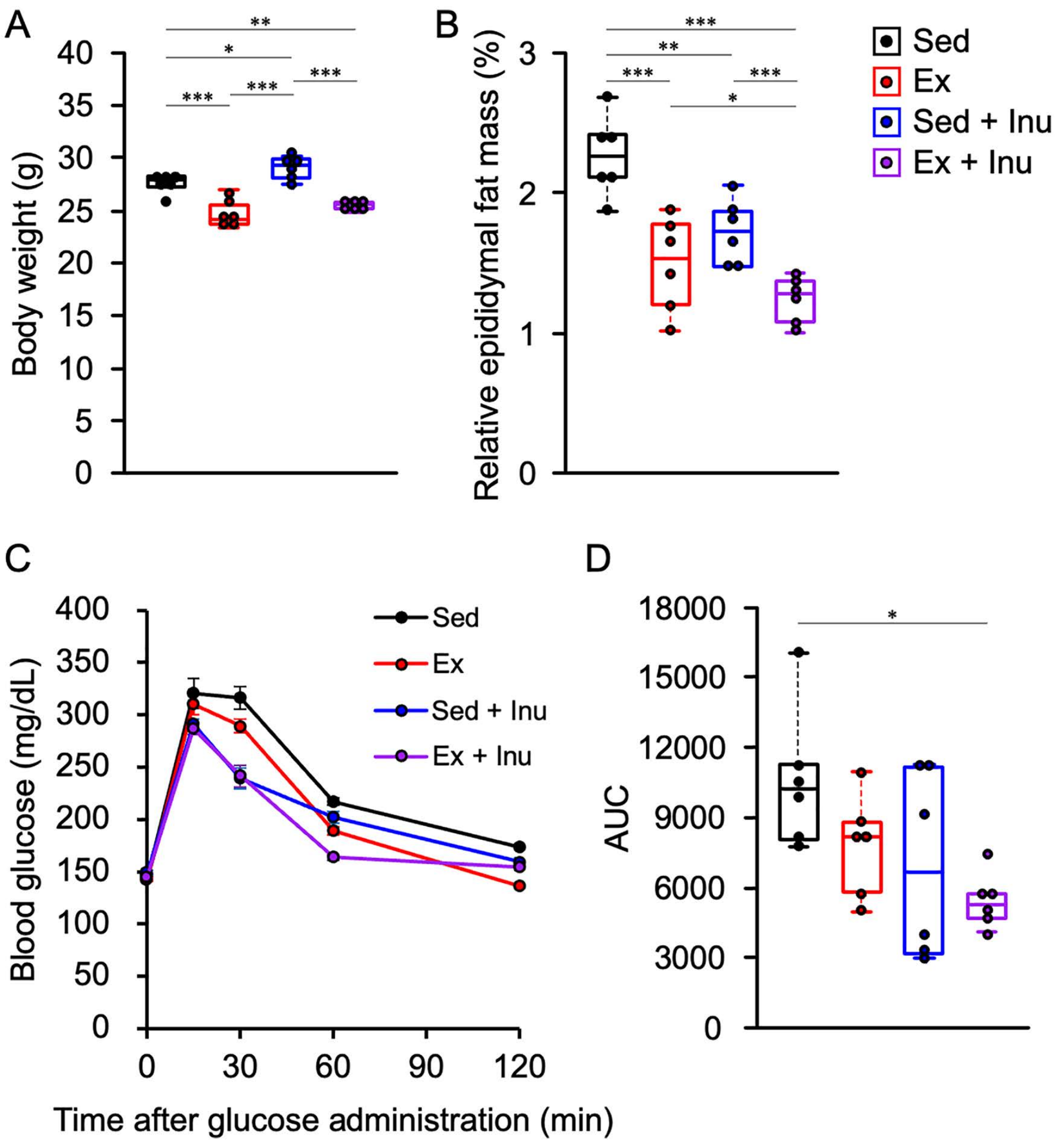


**Fig 2. Physical characteristics and glucose tolerance in donor mice.** **A**: body weight at the end of the feeding. **B**: relative epididymal fat mass to body weight. **C**: blood glucose levels during GTT prior to FMT. **D**: AUC of GTT. Data are expressed as beeswarm boxplots or means ± SEM (n = 6 per group). Statistical significance was assessed with one-way ANOVA followed by Bonferroni's multiple comparison test in **(A)** and **(B)** and with Kruskal-Wallis test followed by Steel-Dwass's post hoc test in **(D)**. *P < 0.05. **P < 0.01. ***P < 0.001. ANOVA: analysis of variance, AUC: area under the curve, FMT: fecal microbiota transplantation, GTT: glucose tolerance test, SEM: standard error of the mean.


and inulin supplementation led to the greatest reduction in relative fat mass, observed in the Ex + Inu group (Fig 2B). In accordance with this, only the combined intervention resulted in a synergistic improvement in glucose metabolism, showing a significant difference compared with the sedentary control (Fig 2C and 2D, P < 0.05).

### Gut microbiota composition in donors and recipients

To assess the composition of bacterial communities in donors and its changes in recipients following FMT, 16S rRNA sequencing using fecal samples was conducted and general landscape of the gut microbiota at the family level was assessed (Fig 3). Taxonomic profiling revealed a distinct microbial composition between donors and recipients prior to FMT (Fig 3). Upon antibiotic treatment, a large decrease of microbial community was noted and *Streptococcaceae* family

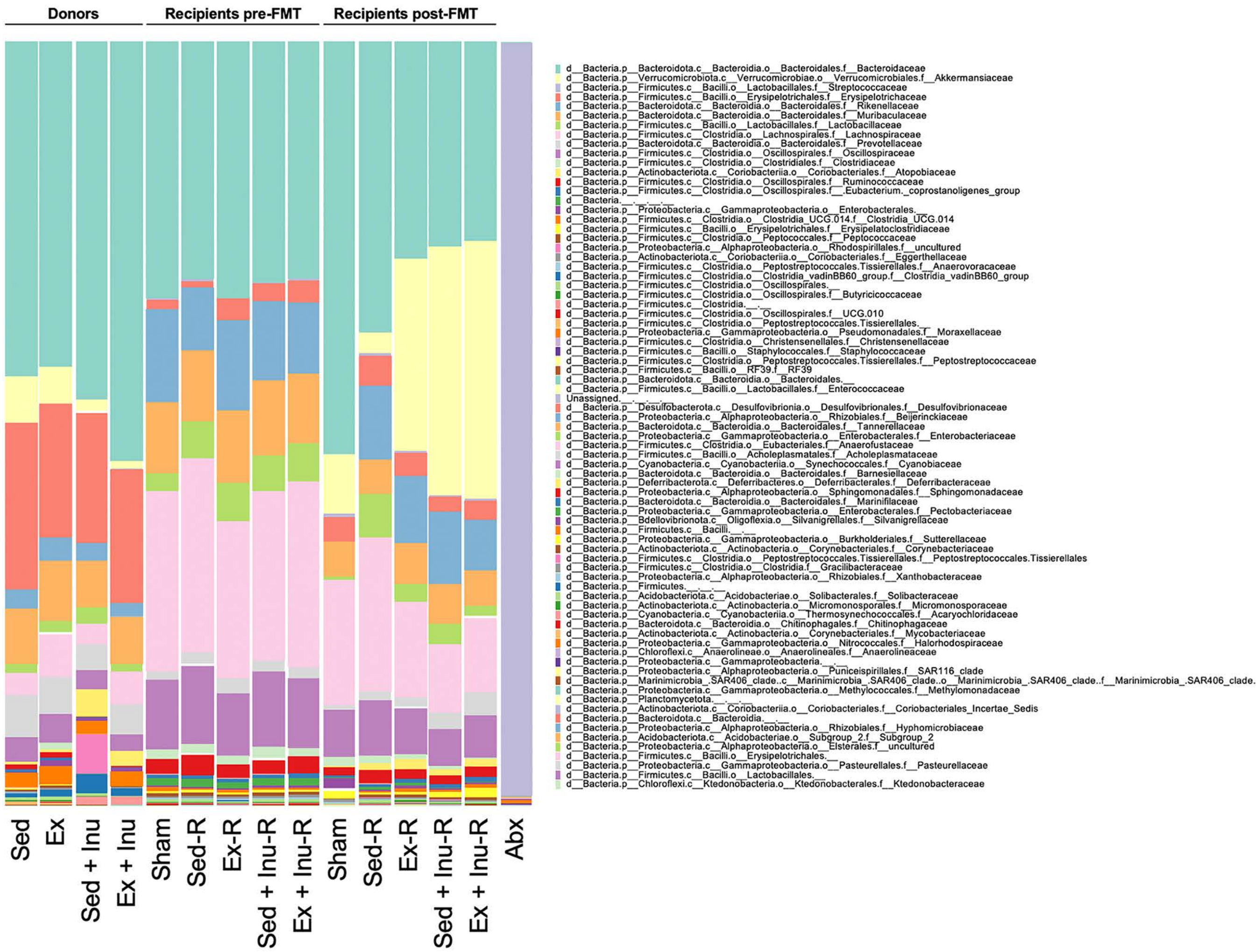


**Fig 3. Relative abundance of the gut microbiota determined using 16S rRNA sequences at the family level.** Fecal samples from donors (Sed, Ex, Sed + Inu, and Ex + Inu), recipients (Sham, Sed-R, Ex-R, Sed + Inu-R, and Ex + Inu-R) at pre- and post-FMT, and following Abx were analyzed. Fecal samples for the Abx group were randomly collected from five recipient mice after antibiotic treatment. The mean values of the relative abundances of gut microbiota taxa in each group are presented. Abx: antibiotics, FMT: fecal microbiota transplantation.


largely dominated in the gut bacterial communities in recipient mice (Abx in Fig 3). Following FMT, gut microbiota in recipient groups were reconstructed with some alterations, particularly a remarkable increase of *Akkermansiaceae* family abundance in the Ex-R, Sed + Inu-R, and Ex + Inu-R groups compared to the Sham and Sed-R groups (Fig 3). However, gut microbiota compositions in recipient mice after FMT remained strikingly different from those in respective donors, suggesting that the HFHSD outperforms FMT for reshaping the gut microbiota after antibiotic treatment (Fig 3).

In the comparison of α-diversity among recipients after FMT, observed ASVs and Chao1 index in the FMT-treated (i.e., Sed-R, Ex-R, Sed + Inu-R, and Ex + Inu-R) groups were significantly higher than those in the Sham group (Fig 4A and 4B, $P < 0.001$). Moreover, in the Sed-R, Ex-R, and Ex + Inu-R groups, the Shannon index was significantly greater than that in the Sham group (Fig 4C, $P < 0.01$). Also, the Shannon index in the Sed + Inu-R group was significantly lower than that in the Sed-R and Ex-R groups (Fig 4C, $P < 0.01$). The difference in the Simpson index among recipient groups was not statistically significant (Fig 4D). In the comparison of β-diversity, PCoA showed that the microbiome composition of the Sham

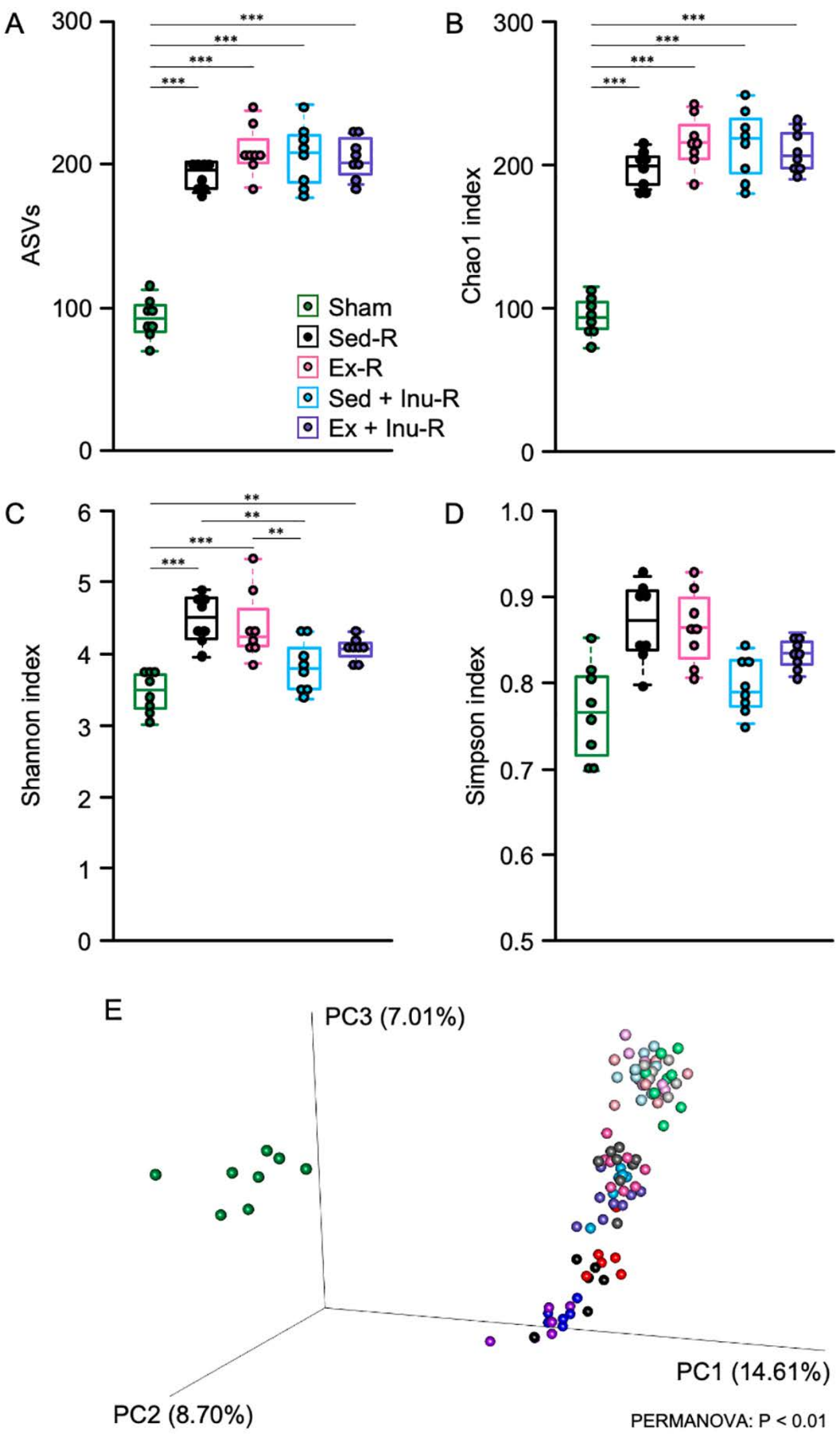


**Fig 4. Bacterial taxonomic signature in donors and recipients. A**–**D**: ASVs **(A)**, Chao1 index **(B)**, Shannon index **(C)**, and Simpson index **(D)** in the fecal samples from recipient mice following FMT. **E**: PCoA based on the unweighted UniFrac distance matrix. Fecal samples from donors (Sed: black, Ex: red, Sed + Inu: blue, and Ex + Inu: purple) and recipients both before (Sham pre: light green, Sed-R pre: light gray, Ex-R pre: light pink, Sed + Inu-R pre: light sky blue, and Ex + Inu-R pre: light purple) and after (Sham post: green, Sed-R post: dark gray, Ex-R post: pink, Sed + Inu-R post: sky blue, and Ex + Inu-R post: light blue violet) FMT were analyzed. Data are expressed as beeswarm boxplots (n = 8 per group). Statistical significance was assessed with one-way ANOVA followed by Bonferroni's multiple comparison test in **(A)**–**(C)**, with Kruskal-Wallis test in **(D)**, and with PERMANOVA in **(E)**. *P < 0.05, **P < 0.01, ***P < 0.001. ANOVA: analysis of variance, ASV: amplicon sequence variant, FMT: fecal microbiota transplantation, PCoA: principal coordinate analysis, PERMANOVA: permutational multivariate analysis of variance.


post-FMT group (green) was clearly separated from those of other recipient groups (Fig 4E). Furthermore, although the gut microbiota composition in recipients shifted toward that of the donors following FMT, it did not cluster with them, and no differences were observed based on FMT type (Fig 4E).

The LEfSe analysis of the gut microbiota in donors displayed the differentially abundant bacteria in the Ex, Sed + Inu, Ex + Inu groups compared with the Sed group (Fig 5A–5C). Unexpectedly, a little difference between the Ex and Sed groups were noted, indicative of a limited modification of gut microbiota by voluntary wheel running exercise in the present

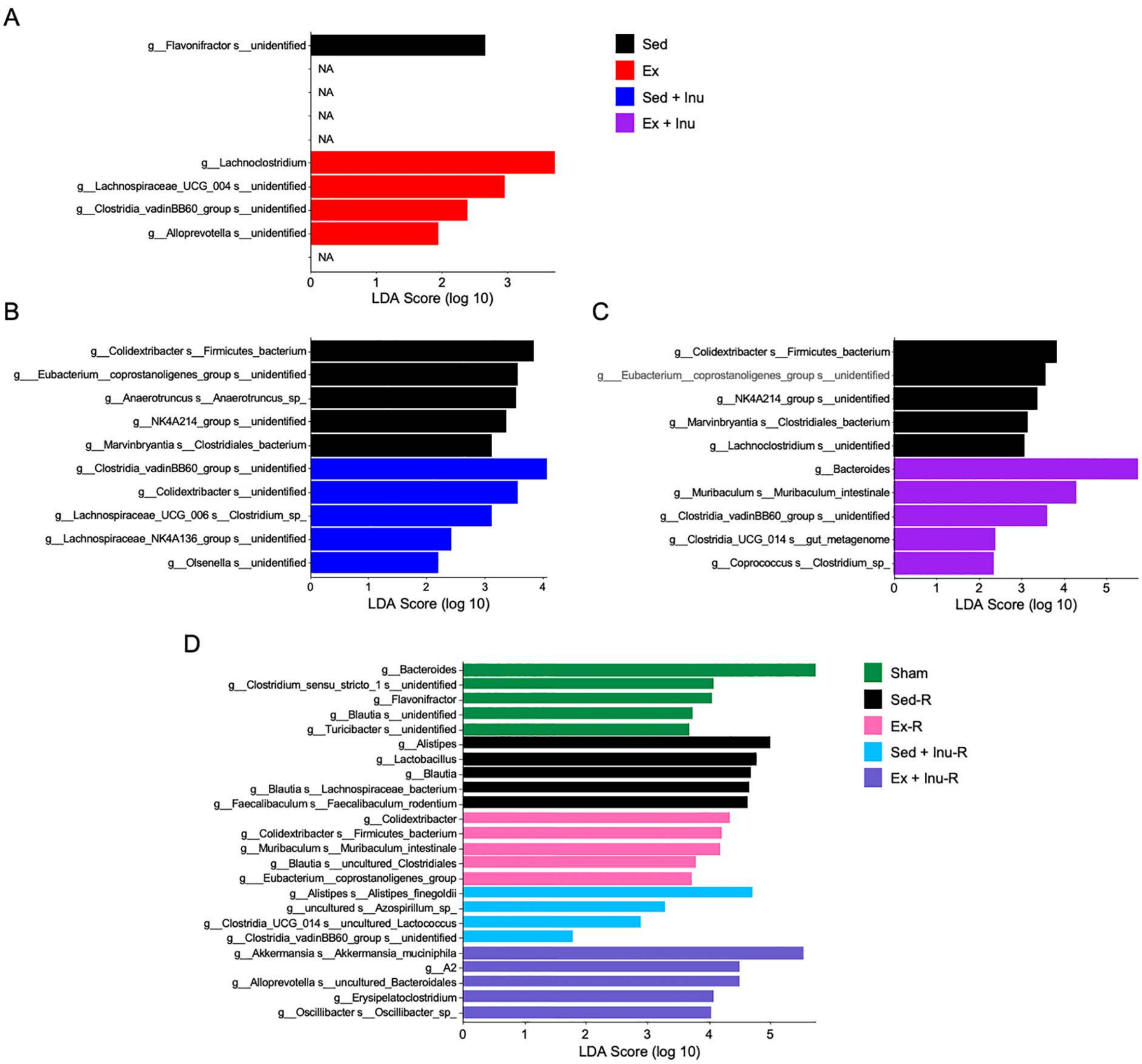


**Fig 5. Linear discriminant analysis (LDA) effect size (LEfSe) analysis of gut microbiota composition in donors and recipients. A**: Gut microbiota stratified by exercise in donors. **B**: Gut microbiota stratified by inulin supplementation in donors. **C**: Gut microbiota stratified by the combination of exercise and inulin supplementation in donors. **D**: Gut microbiota stratified by each FMT in recipients. The top 5 microbiota at taxonomic levels below genus that were significantly more abundant in each group ($P < 0.05$) compared with the other groups are shown. The x-axis indicates the log 10-transformed mean relative abundance of each species. Fecal samples from both donors and recipients after FMT were analyzed. FMT: fecal microbiota transplantation.


study (Fig 5A). Moreover, the LEfSe analysis identified some SCFA producers, including *Lachnospiraceae NK4A136 group* [33] in the Sed + Inu group (Fig 5B) and *Muribaculum intestinale* [34] in the Ex + Inu group (Fig 5C).

In the comparison among recipients, FMT was associated with an increased relative abundance of certain taxa that have been linked to improved metabolic outcomes, such as *Alistipes finegoldii* [35,36] in the Sed + Inu-R group and *Akkermansia muciniphila* [37,38] in the Ex + Inu-R group (Fig 5D). However, these bacteria were not significantly abundant in their respective donors (Fig 5B and 5C).

### Obesity-related parameters in recipient mice

Body weights in all five recipient groups gradually increased during the obesity induction period (Fig 6A). Subsequently, slight reductions in body weight were observed during the 4-week FMT intervention in all groups (Fig 6A); however,

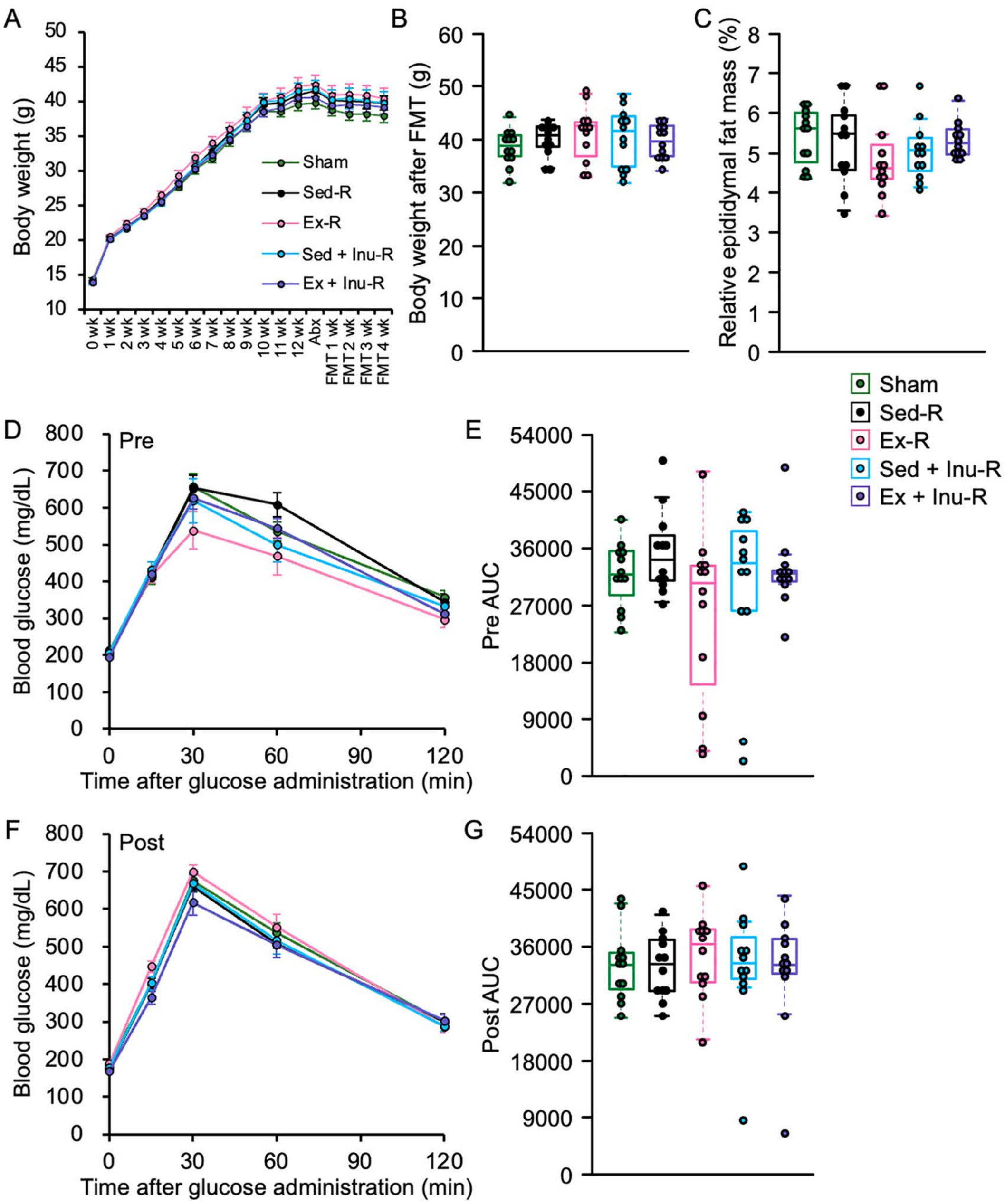


**Fig 6. Physical parameters and glucose metabolism before and after FMT in recipient mice. A**: changes in body weight during the experimental period. **B**: body weight at the end of the feeding. **C**: relative epididymal fat mass to body weight. **D**: blood glucose levels during GTT prior to FMT. **E**: AUC of GTT prior to FMT. **F**: blood glucose levels during GTT following FMT. G: AUC of GTT following FMT. Data are expressed as means ± SEM or beeswarm boxplots (n = 12 per group). Statistical significance was assessed with Kruskal-Wallis test in **(B)**, **(E)**, and **(G)**, and with one-way ANOVA in **(C)**. Abx: antibiotics, ANOVA: analysis of variance, AUC: area under the curve, FMT: fecal microbiota transplantation, GTT: glucose tolerance test, SEM: standard error of the mean.


none of the FMT types elicited significant effects on weight loss or fat mass reduction (Fig 6B and 6C). Consistently, no improvements in glucose tolerance were evident, with glucose levels remaining comparable to pre-FMT values across all groups (Fig 6D–6G). Similarly, no significant differences in circulating lipid parameters or hepatic injury markers were detected among the recipient groups (Fig 7).

### The mRNA expressions in the large intestine of recipient mice

To investigate whether FMT from exercised and/or inulin-fed donors alleviates chronic inflammation in obese recipients, the mRNA expression levels of proinflammatory cytokines, including tumor necrosis factor-alpha (TNF-α) and interleukin-1 beta (IL-1β), were measured in the colonic tissue of recipient mice (Fig 8A and 8B). No significant differences in the expression levels of TNF-α or IL-1β were observed among the recipient groups (Fig 8A and 8B).

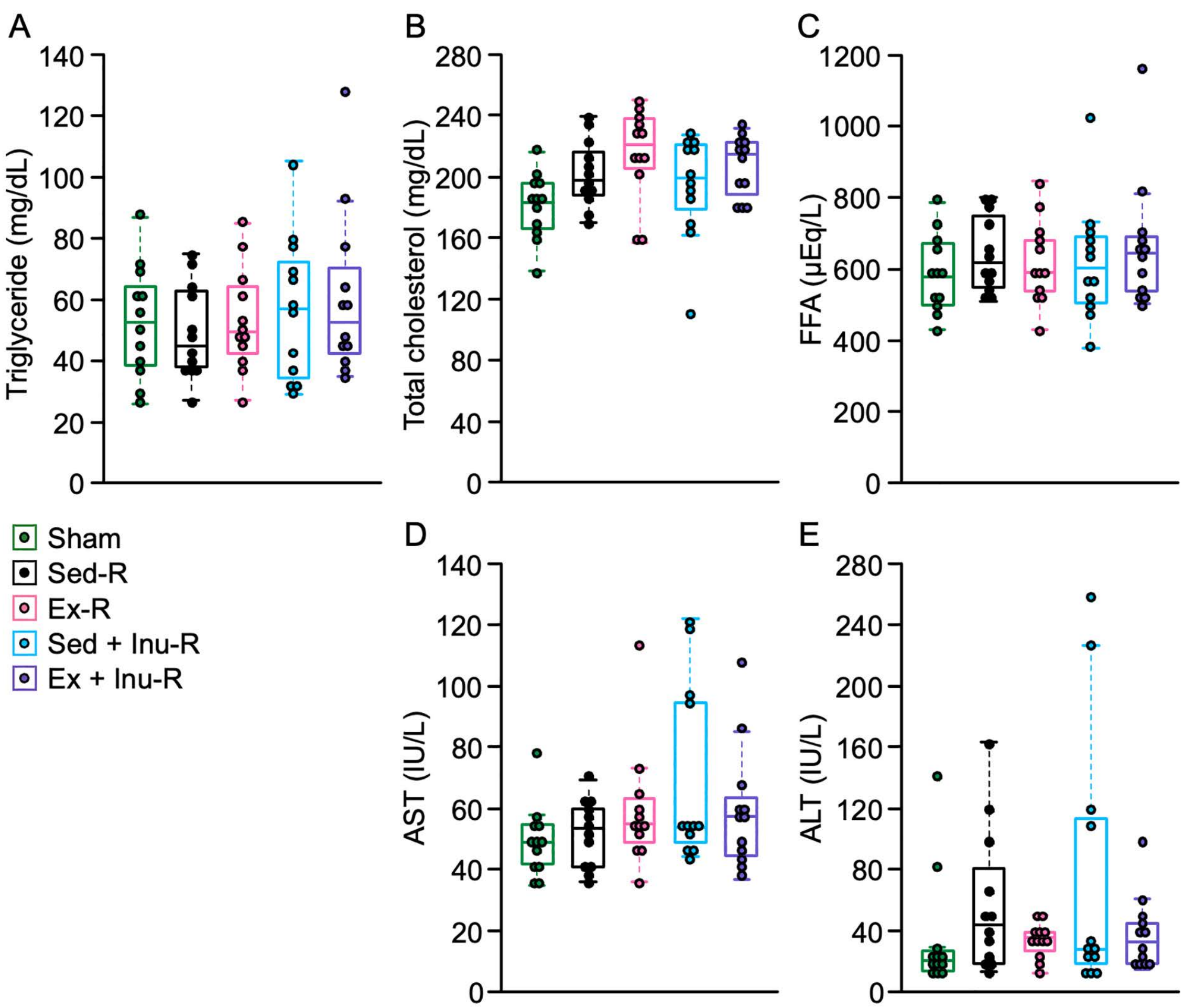


**Fig 7. Circulating metabolic parameters in recipient mice following FMT. A**: triglyceride levels. **B**: total cholesterol levels. **C**: FFA levels. **D**: AST levels. **E**: ALT levels. Data are expressed as beeswarm boxplots (n = 12 per group) and statistical significance was assessed with Kruskal-Wallis test followed by Steel-Dwass's post hoc test. ALT: alanine aminotransferase, AST: aspartate aminotransferase, FFA: free fatty acid, FMT: fecal microbiota transplantation.


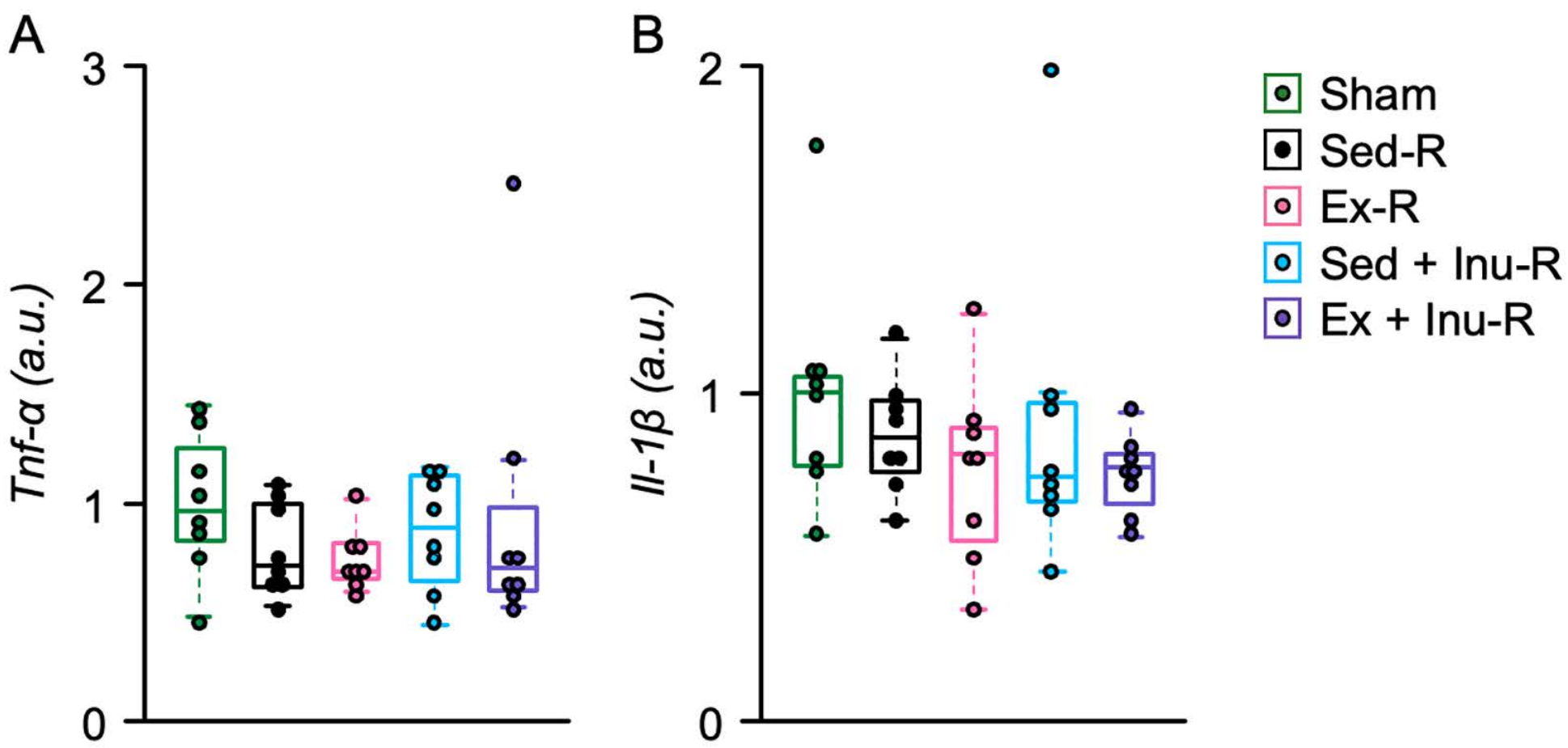


**Fig 8. The mRNA expressions in the large intestine of recipient mice at the end of the experiment. A**: *Tnf-α* mRNA expression levels. **B**: *Il-1β* mRNA expression levels. The mRNA expression levels were normalized to the *Gapdh* used as the internal control and expressed as the fold changes from the mean value in the Sham group. Data are expressed as beeswarm boxplots (n = 8 per group). Statistical significance was assessed with Kruskal-Wallis test. ANOVA: analysis of variance, a.u.: arbitrary unit.


### Cecal SCFA content in donor and recipient mice

At the end of the feeding period, the cecal content of major SCFAs (acetate, propionate, and butyrate) were measured in both donor and recipient mice (Fig 9). In the donor groups, the levels of acetate and butyrate were significantly increased in the inulin-supplemented (Sed + Inu and Ex + Inu) groups compared with the Sed and Ex groups ($P < 0.05$), which did not consume inulin (Fig 9A and 9C). Moreover, the acetate level in the Ex + Inu group was significantly lower than that in the Sed + Inu group (Fig 9A, $P < 0.05$). Additionally, the propionate level was significantly higher in the Sed + Inu group than in the Ex group (Fig 9B, $P < 0.05$).

In the recipient groups, no significant difference in acetate levels was observed among the groups (Fig 9D). However, propionate levels were significantly higher in the Ex-R and Sed + Inu-R groups than in the Sham group (Fig 9E, $P < 0.05$ for both). Furthermore, propionate levels were significantly higher in the Ex-R group than in the Sed-R group (Fig 9E, $P < 0.05$). Unexpectedly, the butyrate level was significantly lower in the Ex + Inu-R group than in the Sham, Sed-R, and Ex-R groups (Fig 9F, $P < 0.01$).

## Discussion

The present study examined the effects of FMT from donor mice subjected to voluntary exercise and/or inulin supplementation on obesity and its associated metabolic disorders in HFHSD-fed obese recipient mice. The results demonstrated that exercise and/or inulin supplementation in donor mice led to reduced fat mass, and their combination synergistically improved glucose homeostasis (Fig 2). FMT from these donors altered gut microbial composition (Figs 3–5) and increased cecal propionate levels (Fig 9E) in obese recipients. However, no beneficial effects of FMT on obesity-related parameters (Figs 6A–6C and 7), glucose metabolism (Fig 6D–6G), or colonic inflammatory markers (Fig 8) were observed in the recipients. Thus, the present study revealed that the metabolic benefits of exercise and/or inulin in donors may not be transmissible via FMT to HFHSD-fed obese recipients.

In the present study, the 4-week-FMT from exercised and/or inulin-supplemented donors failed to ameliorate physical parameters (Fig 6A–6C) or glucose tolerance (Fig 6D–6G) in obese recipients. In contrast, previous studies have reported significant improvements in obesity-related outcomes following FMT from exercised [39] or inulin-fed [40] donors. For

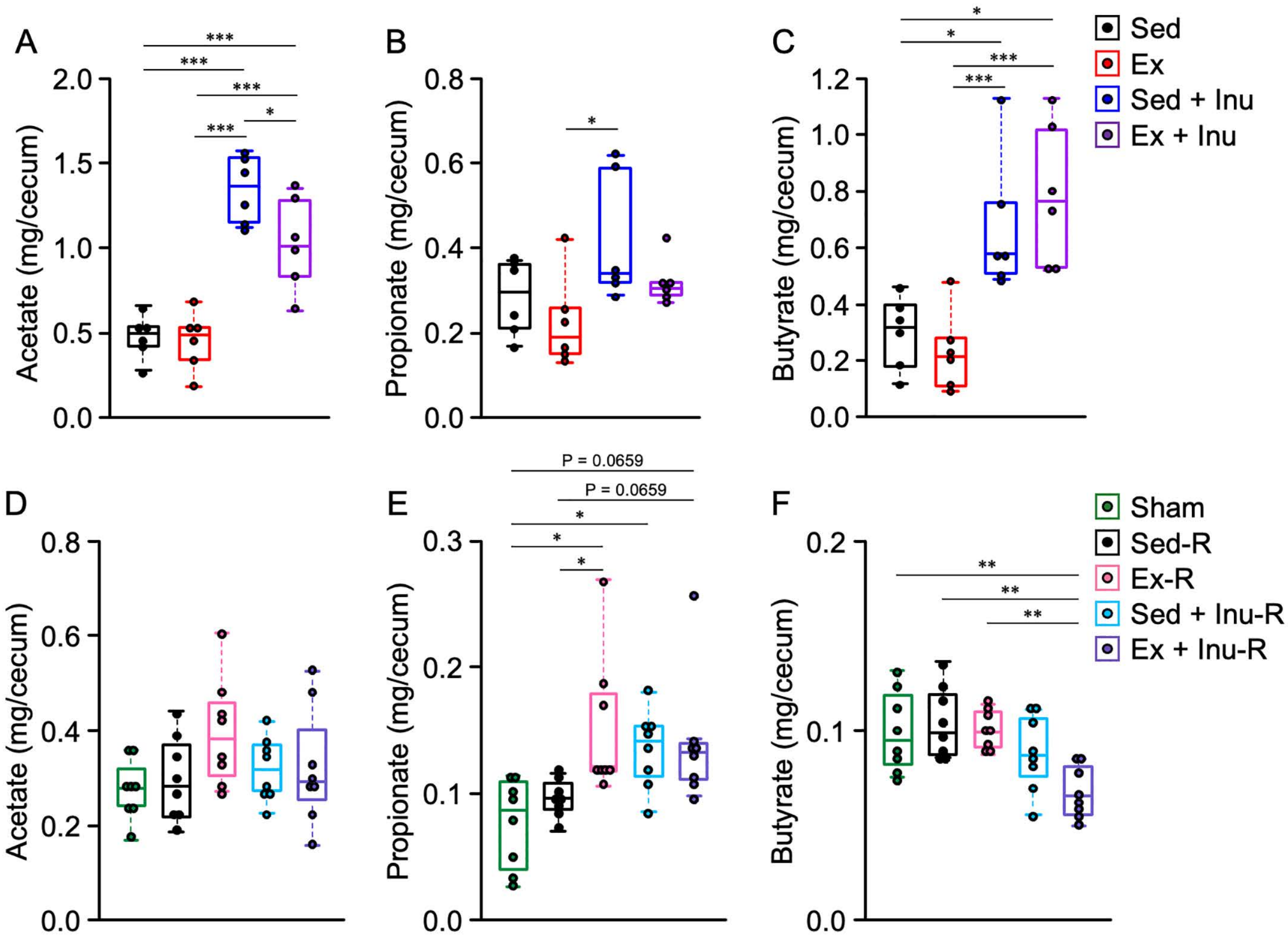


**Fig 9. Cecal SCFA contents in donor and recipient mice at the end of the experiment. A**–**C**: acetate **(A)**, propionate **(B)**, and butyrate **(C)** levels in the cecum of donor mice. **D**–**F**: acetate **(D)**, propionate **(E)**, and butyrate **(F)** levels in the cecum of recipient mice. Data are expressed as beeswarm box-plots (n = 8 per group). Statistical significance was assessed with one-way ANOVA followed by Bonferroni's multiple comparison test in **(A)**, **(D)**, and **(F)**, and with Kruskal-Wallis test followed by Steel-Dwass's post hoc test in **(B)**, **(C)**, and **(E)**. *P < 0.05. **P < 0.01. ***P < 0.001. ANOVA: analysis of variance, SCFA: short-chain fatty acid.


instance, Lai et al. (2018) administered a 12-week FMT from exercised donors to HFD-fed obese mice, resulting in suppressed weight gain, reduced fat mass, and improved glucose metabolism, likely due to the successful transfer of beneficial genera such as *Odoribacter* and *Helicobacter* [39]. In this previous study, however, recipient mice underwent 7 weeks of HFD feeding and weighed only 25–27 g at the beginning of FMT, reflecting an early stage of obesity [39]. In contrast, in the present study, the average body weight of recipient mice exceeded 40 g at FMT initiation (Fig 6A), indicative of more advanced obesity. These discrepancies suggest that FMT from exercised donors may exert preventive, rather than therapeutic, effects on obesity and associated metabolic dysfunction. Regarding inulin, beneficial effects of inulin intake in body composition, glucose homeostasis, and systemic inflammation in obese donors were successfully transferred via a 2-week FMT to recipient mice previously subjected to 10 weeks of HFD [40]. These effects were accompanied by successful transplantation of genera such as *Bifidobacterium* and *Muribaculum* [40]. In the present study, although some beneficial bacteria, including *Muribaculum intestinale* and *A. muciniphila*, were detected in recipient feces (Fig 5D), the

dominant microbiota of exercised and/or inulin-fed donors was not recapitulated in recipients (Fig 5). A major distinction is that our recipient mice were continuously exposed to a combination of HFD and sugary water (42 g/L; 55% fructose and 45% sucrose), which is known to exacerbate obesity-related MASLD (formerly non-alcoholic fatty liver disease: NAFLD) progression [18]. This persistent dietary challenge throughout the experimental period, including FMT, might impair the engraftment of donor-derived microbes following antibiotic treatment. Indeed, the previous study [39] emphasized that microbial colonization post-FMT depends on the recipient's diet. Collectively, it is plausible that the gut environment under continuous HFHSD exposure overrode the beneficial microbial signals from exercised and/or inulin-supplemented donors, limiting the capacity of FMT to induce metabolic improvements.

Accumulating evidence has highlighted the therapeutic potential of FMT in improving host metabolism across both animal models and human subjects [8,39–42]. In the present study, although no overt metabolic benefits of FMT were observed, the FMT from exercised and/or inulin-supplemented donors led to an increase in certain beneficial bacterial taxa in recipient feces, such as *A. muciniphila* in the Ex + Inu-R group and *A. finegoldii* in the Sed + Inu-R group (Fig 5D). These species are recognized SCFA producers [43,44] and have been implicated in ameliorating obesity and related metabolic dysfunctions [35–38]. However, their roles in metabolic regulation may be context-dependent. For instance, *A. muciniphila* was found to increase following Roux-en-Y gastric bypass surgery in patients with severe obesity, yet this change was not correlated with improved metabolic outcomes [45]. Similarly, although *A. finegoldii* was enriched in obese individuals, its abundance decreased in parallel with improvements in metabolic parameters [46]. These inconsistent findings suggest that microbial effects may be context-dependent under dysbiotic conditions. Additionally, the FMT intervention resulted in limited increases in cecal propionate contents (Fig 9E), but had no impact on other major SCFAs (Fig 9D and 9F). Notably, all SCFA levels in recipients remained less than half of those observed in donors (Fig 9), suggesting that the transferred microbiota may not have sufficiently restored SCFA production to levels associated with anti-obesity effects [47,48]. These observations are consistent with the limited metabolic impact of FMT observed in the present study, although this interpretation should be made with caution given the limited functional resolution of 16S rRNA-based microbiome analysis.

There are several limitations to consider when interpreting the findings of the present study. First, although FMT altered gut microbial composition in recipients (Figs 3–5), the extent of donor-derived microbial engraftment remains unclear. It is important to distinguish between failed engraftment and successful engraftment with suppressed microbial functionality, as host environmental factors can constrain microbial activity even when colonization occurs [41,49]. Second, the FMT protocol did not include quantitative assessments of microbial load, viability, or absolute bacterial abundance in donor inocula or recipient feces. In addition, although antibiotic treatment markedly altered gut microbial composition (Fig 3), as previously reported [22], the extent of bacterial depletion was not quantitatively assessed. These parameters are increasingly recognized as critical determinants of FMT efficacy and reproducibility [50]. Third, the incomplete recapitulation of donor microbial signatures in recipients may reflect not only dietary effects but also methodological factors, including pooling of donor fecal samples, variability in sample preparation, and loss of obligate anaerobes during processing [50,51]. Finally, the exclusive use of male mice limits generalizability, given known sex-dependent differences in gut microbiota composition and metabolic responses [52]. Taken together, these limitations highlight the need for more comprehensive and mechanistic approaches to better understand the determinants of FMT efficacy in obesity and metabolic disorders.

In summary, the present study suggests that metabolic benefits conferred by habitual exercise and/or inulin supplementation in lean donors were not readily transmitted via FMT to obese recipient mice, persistently fed with HFHSD. This was accompanied by a mismatch in gut microbiota composition between donors and recipients. These findings highlight the importance of optimizing the host intestinal environment—through dietary modification or microbiome-targeted strategies—to support the colonization and functional activity of beneficial microbes for effective obesity management.

## Supporting information

**S1 Appendix. Underlying data for Figs 2–9.**
(ZIP)

## Acknowledgments

The authors would like to express the members of our laboratory for their cooperation.

## Author contributions

**Conceptualization:** Masato Kawashima, Takafumi Aoki, Hiromi Yano.

**Formal analysis:** Masato Kawashima, Takafumi Aoki, Hiroki Hamada.

**Funding acquisition:** Masato Kawashima.

**Investigation:** Masato Kawashima, Takafumi Aoki, Hiroki Hamada, Chihiro Watanabe, Eri Oyanagi, Takumi Kono, Takashi Yamagata, Hiromi Yano.

**Methodology:** Masato Kawashima, Takafumi Aoki, Eri Oyanagi, Hiromi Yano.

**Project administration:** Masato Kawashima.

**Supervision:** Nicholas P West, Hiromi Yano.

**Visualization:** Masato Kawashima, Hiroki Hamada.

**Writing – original draft:** Masato Kawashima.

**Writing – review & editing:** Takafumi Aoki, Hiroki Hamada, Chihiro Watanabe, Eri Oyanagi, Takumi Kono, Takashi Yamagata, Nicholas P West, Hiromi Yano.

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
