## [Decision Letter · Decision Letter 0]

18 Mar 2026

PONE-D-25-44891Continuous high-fat high-sugar diet overrides the therapeutic potential of fecal microbiota transplantation from exercised and/or inulin-conditioned donors in obese micePLOS One

Dear Dr. Kawashima,

Thank you for submitting your manuscript to PLOS ONE. After careful consideration, we feel that it has merit but does not fully meet PLOS ONE’s publication criteria as it currently stands. Therefore, we invite you to submit a revised version of the manuscript that addresses the points raised during the review process.

We look forward to receiving your revised manuscript.

Kind regards,

Daniel Paredes-Sabja, PhD

Academic Editor

PLOS One

Journal Requirements:

3. Thank you for stating the following financial disclosure: [This study was supported by JSPS KAKENHI Grant Number JP21K21270 (to M.K.) (URL: https://kaken.nii.ac.jp/en/grant/KAKENHI-PROJECT-21K21270/) and JP23K16738 (to M.K.) (URL: https://kaken.nii.ac.jp/en/grant/KAKENHI-PROJECT-23K16738/), and The Ryobiteien Memorial Foundation (URL: https://ryobi.gr.jp/teienzaidan/index.html) (to M.K.).].

4. Please expand the acronym “JSPS” (as indicated in your financial disclosure) so that it states the name of your funders in full.

5. We note that your Data Availability Statement is currently as follows: [All relevant data are within the manuscript and its Supporting Information files.]

Additional Editor Comments:

I have acted as a reviewer myself and the critics are as follow:

Major weaknesses:

1. Experimental design limits interpretability but remains scientifically valid. Can the authors further discuss failed engraftment and successful engraftment with suppressed functionality. This distinction should be more explicitly acknowledged in the Discussion.

2. FMT protocol lacks quantitative validation. The study does not report microbial load, viability, or absolute bacterial counts in donor suspensions or recipient feces post-FMT. This should be discusssed as well.

3. Microbiome analysis is descriptive given that authors conducted 16S rRNA sequencing. Hence conclusions and statements linking SCFA levels or specific taxa to host metabolic outcomes should be more cautiously framed.

4. Antibiotic depletion efficiency not quantified. How do the authors ensure proper bacterial depletion. Needs to be addressed.

5. The manuscript emphasizes that recipients did not recapitulate donor microbial signatures, and does not consider explanations such as pooling effects, fecal preparation variability, or loss of anaerobic conditions during processing.

Minor weaknesses

1. Sex limitation not adequately discussed, as authors exclusively used male mice.

2. Statistical reporting could be clarified. For example, results described as “trend-level” (P ≈ 0.06) are interpreted biologically. These should either be clearly labeled as non-significant or omitted from statements.

3. LEfSe results seem to undergo overinterpretation; The authors should clarify that these findings are associative rather than causal.

5. Redundancy in discussion as in several sections authors reiterate results already described in detail, particularly regarding diet dominance over FMT. This could be streamlined for clarity.

6. PLOS authors have the option to publish the peer review history of their article (what does this mean?). If published, this will include your full peer review and any attached files.

---

## [Author Response · Author response to Decision Letter 1]

19 Apr 2026

We wish to express our appreciation to reviewers for his or her insightful comments, which have helped us significantly improve the paper. Our responses are as follows.

Major comments

Q1. Experimental design limits interpretability but remains scientifically valid. Can the authors further discuss failed engraftment and successful engraftment with suppressed functionality. This distinction should be more explicitly acknowledged in the Discussion.

A1. We thank the reviewer for this important comment. We agree that distinguishing between failed engraftment and successful engraftment with suppressed functionality is critical for interpreting our findings. In the revised manuscript, we have expanded the Discussion to explicitly address this point. We now discuss the possibility that donor-derived microbes may have colonized the recipient gut but failed to exert metabolic effects due to the host intestinal environment under continuous HFHSD feeding. This interpretation is supported by previous studies demonstrating that host-specific factors can limit microbial engraftment and function following FMT (Kootte et al., Cell Metab, 2017; Zmora et al., Cell, 2018). This point has been incorporated into the newly added limitation paragraph in the Discussion section (page 22, lines 413-417; page 23, line 418) with following references.

41. Kootte RS, Levin E, Salojärvi J, Smits LP, Hartstra AV, Udayappan SD, et al. Improvement of insulin sensitivity after lean donor feces in metabolic syndrome is driven by baseline intestinal microbiota composition. Cell Metab. 2017;26(4):611–619.e6. doi:10.1016/j.cmet.2017.09.008.

49. Zmora N, Zilberman-Schapira G, Suez J, Mor U, Dori-Bachash M, Bashiardes S, et al. Personalized gut mucosal colonization resistance to empiric probiotics is associated with unique host and microbiome features. Cell. 2018;174(6):1388–1405.e21. doi:10.1016/j.cell.2018.08.041.

Q2. FMT protocol lacks quantitative validation. The study does not report microbial load, viability, or absolute bacterial counts in donor suspensions or recipient feces post-FMT. This should be discussed as well.

A2. We appreciate the reviewer’s insightful suggestion. We agree that quantitative validation of the FMT protocol is important. In the revised manuscript, we have added a statement in the Discussion clarifying that microbial load, viability, and absolute bacterial abundance were not quantitatively assessed in donor inocula or recipient feces. These parameters are increasingly recognized as critical determinants of FMT efficacy and reproducibility (Bokoliya et al., Front Cell Infect Microbiol, 2021). This point has now been clearly described in the newly added limitation paragraph in the Discussion section (page 23, lines 418-423) with a following reference.

50. Bokoliya SC, Dorsett Y, Panier H, Zhou Y. Procedures for fecal microbiota transplantation in murine microbiome studies. Front Cell Infect Microbiol. 2021;11:711055. doi:10.3389/fcimb.2021.711055.

Q3. Microbiome analysis is descriptive given that authors conducted 16S rRNA sequencing. Hence conclusions and statements linking SCFA levels or specific taxa to host metabolic outcomes should be more cautiously framed.

A3. We thank the reviewer for this valuable comment. We agree that 16S rRNA sequencing primarily provides taxonomic (descriptive) information and does not allow direct functional inference. In response, we have revised the manuscript to adopt more cautious language when discussing associations between specific microbial taxa, SCFA levels, and host metabolic outcomes (page 22, lines 404-405 and 407-410; page 23, lines 431-433 and 434-437). Moreover, we have added a statement to acknowledge the limited functional resolution of 16S rRNA-based microbiome analysis (page 22, lines 411-412). These revisions ensure that our conclusions are appropriately framed and consistent with the limitations of the methodology.

Q4. Antibiotic depletion efficiency not quantified. How do the authors ensure proper bacterial depletion. Needs to be addressed.

A4. We thank the reviewer for this important comment. We acknowledge that the present study did not include direct quantitative assessment of bacterial depletion efficiency, such as total bacterial load or absolute bacterial abundance following antibiotic treatment. However, as shown in Fig. 3, antibiotic treatment resulted in a marked shift in gut microbial composition, characterized by reduced diversity and the dominance of specific taxa (e.g., Streptococcaceae). Such compositional shifts are consistent with a previous study demonstrating substantial disruption of gut microbiota following broad-spectrum antibiotic treatment (Vijay-Kumar et al., Science, 2010), although these approaches do not provide quantitative assessment of bacterial depletion. Nevertheless, we agree that compositional changes alone are insufficient to fully confirm depletion efficiency. Therefore, we have incorporated this point into the newly added limitation paragraph in the Discussion section (page 23, lines 419-423).

Q5. The manuscript emphasizes that recipients did not recapitulate donor microbial signatures, and does not consider explanations such as pooling effects, fecal preparation variability, or loss of anaerobic conditions during processing.

A5. We thank the reviewer for this valuable suggestion. We agree that methodological factors may have contributed to the incomplete recapitulation of donor microbial signatures in recipients. In the revised manuscript, we have expanded the Discussion to include alternative explanations beyond dietary effects. Specifically, we now discuss the potential influence of pooling of donor fecal samples, variability in fecal preparation procedures, and the loss of obligate anaerobic bacteria during sample handling and processing. These factors have been reported to affect microbial viability and engraftment efficiency in FMT studies (Moayyedi et al., Gastroenterology, 2015; Bokoliya et al., Front Cell Infect Microbiol, 2021). These additions have been incorporated into the newly added limitation paragraph in the Discussion section (page 23, lines 423-426) with following references.

50. Bokoliya SC, Dorsett Y, Panier H, Zhou Y. Procedures for fecal microbiota transplantation in murine microbiome studies. Front Cell Infect Microbiol. 2021;11:711055. doi:10.3389/fcimb.2021.711055.

51. Moayyedi P, Surette MG, Kim PT, Libertucci J, Wolfe M, Onischi C, et al. Fecal microbiota transplantation induces remission in patients with active ulcerative colitis in a randomized controlled trial. Gastroenterology. 2015;149(1):102–109.e6. doi:10.1053/j.gastro.2015.04.001.

Minor comments

Q1. Sex limitation not adequately discussed, as authors exclusively used male mice.

A1. We thank the reviewer for highlighting this important point. We acknowledge that the exclusive use of male mice limits the generalizability of our findings. In the revised manuscript, we have added a statement in the Discussion noting that sex-dependent differences in gut microbiota composition and host metabolic responses have been reported. For example, a previous study has demonstrated that sex is a key determinant of gut microbial structure and function (Org et al., Gut Microbes, 2016). This limitation has now been clearly addressed in the newly added limitation paragraph in the Discussion section (page 23, lines 426-428) with a following reference.

52. Org E, Mehrabian M, Parks BW, Shipkova P, Liu X, Drake TA, et al. Sex differences and hormonal effects on gut microbiota composition in mice. Gut Microbes. 2016;7(4):313–322. doi:10.1080/19490976.2016.1203502.

Q2. Statistical reporting could be clarified. For example, results described as “trend-level” (P ≈ 0.06) are interpreted biologically. These should either be clearly labeled as non-significant or omitted from statements.

A2. We thank the reviewer for this pertinent comment. We agree that results with P values above the significance threshold should not be interpreted as biologically meaningful. In the revised manuscript, we have removed all “trend-level” and “tended to” expressions and clarified that results with P ≥ 0.05 are considered non-significant. Corresponding statements in both the Results (page 19, lines 340-342) and Discussion (page 22, lines 405-407) sections have been revised accordingly to avoid overinterpretation.

Q3. LEfSe results seem to undergo overinterpretation; The authors should clarify that these findings are associative rather than causal.

A3. We thank the reviewer for this important comment. We agree that LEfSe analysis based on 16S rRNA sequencing provides associative rather than causal insights. In the revised manuscript, we have modified the relevant descriptions in the Results section (page 15, lines 276-277; page 16, lines 278-279) to avoid causal interpretation and to more clearly reflect associative relationships.

Q4. Redundancy in discussion as in several sections authors reiterate results already described in detail, particularly regarding diet dominance over FMT. This could be streamlined for clarity.

A4. We thank the reviewer for this helpful comment. We acknowledge that parts of the Discussion contained redundant descriptions, particularly regarding the dominant effects of diet over FMT. In the revised manuscript, we have streamlined the Discussion by removing repetitive statements (page 22, lines 404-405 and 410-412) and consolidating the key message.

We appreciate the helpful suggestions offered by expert reviewers as their comments were extremely useful for revising this manuscript. We declare that this manuscript is not simultaneously submitted to other journals. We would be grateful if the manuscript could be reviewed and considered for publication in PLoS One as a research article.

---

## [Editor Report · Decision Letter 1]

28 Apr 2026

Continuous high-fat high-sugar diet overrides the therapeutic potential of fecal microbiota transplantation from exercised and/or inulin-conditioned donors in obese mice

PONE-D-25-44891R1

Dear Dr. Kawashima,

We’re pleased to inform you that your manuscript has been judged scientifically suitable for publication and will be formally accepted for publication once it meets all outstanding technical requirements.

Kind regards,

Daniel Paredes-Sabja, PhD

Academic Editor

PLOS One
---

## [Editor Report · Acceptance letter]

PONE-D-25-44891R1

PLOS One

Dear Dr. Kawashima,

I'm pleased to inform you that your manuscript has been deemed suitable for publication in PLOS One. Congratulations! Your manuscript is now being handed over to our production team.

Kind regards,

on behalf of

Dr. Daniel Paredes-Sabja

Academic Editor

PLOS One